# Molecular Dynamics Study of Clathrate-like Ordering of Water in Supersaturated Methane Solution at Low Pressure

**DOI:** 10.3390/molecules28072960

**Published:** 2023-03-26

**Authors:** Rodion V. Belosludov, Kirill V. Gets, Ravil K. Zhdanov, Yulia Y. Bozhko, Vladimir R. Belosludov, Li-Jen Chen, Yoshiyuki Kawazoe

**Affiliations:** 1Institute for Materials Research, Tohoku University, Sendai 980-8577, Japan; 2Nikolaev Institute of Inorganic Chemistry, Siberian Branch, Russian Academy of Sciences, 630090 Novosibirsk, Russia; 3Department of Physics, Novosibirsk State University, 630090 Novosibirsk, Russia; 4Department of Chemical Engineering, National Taiwan University, Taipei 10617, Taiwan; 5New Industry Creation Hatchery Center, Tohoku University, Sendai 980-8579, Japan; 6Department of Physics and Nanotechnology, SRM Institute of Science and Technology, Kattankurathur 603203, India; 7School of Physics, Institute of Science, Suranaree University of Technology, Nakhon Ratchasima 30000, Thailand

**Keywords:** gas hydrates, nucleation mechanism, amorphous hydrate, intramolecular hydrogen bonds, computer simulation

## Abstract

Using molecular dynamics, the evolution of a metastable solution for “methane + water” was studied for concentrations of 3.36, 6.5, 9.45, 12.2, and 14.8 mol% methane at 270 K and 1 bar during 100 ns. We have found the intriguing behavior of the system containing over 10,000 water molecules: the formation of hydrate-like structures is observed at 6.5 and 9.45 mol% concentrations throughout the entire solution volume. This formation of “blobs” and the following amorphous hydrate were studied. The creation of a metastable methane solution through supersaturation is the key to triggering the collective process of hydrate formation under low pressure. Even the first stage (0–1 ns), before the first fluctuating cavities appear, is a collective process of H-bond network reorganization. The formation of fluctuation cavities appears before steady hydrate growth begins and is associated with a preceding uniform increase in the water molecule’s tetrahedrality. Later, the constantly presented hydrate cavities become the foundation for a few independent hydrate nucleation centers, this evolution is consistent with the labile cluster and local structure hypotheses. This new mechanism of hydrogen-bond network reorganization depends on the entropy of the cavity arrangement of the guest molecules in the hydrate lattice and leads to hydrate growth.

## 1. Introduction

Gas hydrates are crystalline compounds with hydrogen-bonded water molecules acting as the host network and guest molecules placed in the host cavities. Typically, for gas hydrate formation, high pressure and low temperature are required. The most common hydrate structures in nature are the cubic structure I (sI), the cubic structure II (sII), and the hexagonal structure H (sH). The formation of the different hydrate structures depends on the size of the guest molecule [1].

Gas hydrate formation is an important process for a number of different applications, such as gas separation, energy storage, energy transport, cold energy storage, sequestration, and desalination. Hydrate formation from the liquid phase involves the crucial step of hydrate nucleation, which is very important for developing efficient hydrates for practical applications [2].

Hydrophobic hydration is important for understanding the dissolution of hydrophobic molecules and the formation of clathrate hydrate. The solubility of hydrophobic molecules in water under normal conditions is very low and does not affect the structure of the solution as a whole but changes it locally. For the formation of a hydrate, it is necessary to dissolve several orders of magnitude of additional molecules. In this case, it is not a local effect; as a result of hydrophobic hydration, the entire structure of the solution is transformed.

The typical time evolution of gas consumption during the hydrate formation process can be divided into three main phases: the dissolution phase, the metastable supersaturated phase, and the growth phase [2]. The dissolution phase starts with increasing gas uptake due to the dissolution into the liquid phase.

Experimental molecular-level description of the hydrate nucleation process in water is challenging not only due to the extremely small length/time scale involved but also due to the stochastic nature of the nucleation process, which is only possible with state-of-the-art measurements. It should also be noted that the anomalous behavior of water is also important. Density fluctuations of water with dimensions of ~0.6 nm have been observed [3] and later confirmed by theoretical investigations [4]. The properties of water are determined by its dynamic hydrogen-bond network [5], which influences the stochastic nature of the process of hydrate nucleation.

Over the past few decades numerous theoretical molecular dynamics (MD) simulations have been performed in which the authors unraveled crucial aspects of crystal nucleation in liquids (see reviews [2,6,7]). These MD simulations have been used to obtain a detailed understanding of the nucleation process in different systems, where the time required for the evolution of a liquid into a crystal is more or less accurately reproduced.

One hypothesis for hydrate formation [1] involves hydrate precursor structures, in which the water molecules form individual clathrate cages (hydrate precursors) around the dissolved guest molecules prior to the hydrate formation process. Then, if favorable conditions are met, these cages will agglomerate and form a solid clathrate hydrate. Using MD simulations of the hydration of hydrophobic molecules, volumetric numerical data describing the details of the process was obtained. The creation of structured hydration shells around the dissolved hydrophobic molecules was shown [8,9,10,11,12,13,14,15,16,17,18,19]. These clusters (often called “postcursors”) are preserved for an extended period of time in the liquid phase [20]. The degree of supercooling influences these shells, and more amorphous structures are formed at lower temperatures [21]. Conversely, in other studies [22,23] the authors did not mention any enhanced structure around the hydrophobic solutes and did not find clathrate cages in solutions with long lifetimes. Experimental studies were carried out to reveal the water structure around non-polar solutes [24,25,26,27,28]. These results are consistent with previous computer simulation data, where there are loosely organized hydration cages around the hydrophobic solutes but without evidence of the regular clathrate hydrate cages in solution that are present in water. For example, an NMR investigation of tetrahydrofuran (THF) hydrate formation observed THF hydration shells with no evidence of hydrate precursor formation [29].

There have been many research studies of the mechanism of gas hydrate nucleation [30,31,32,33,34,35,36]. Currently, there are two main molecular mechanisms for hydrate nucleation. The first is the labile cluster hypothesis (LCH) [30], and the second is the local structure hypothesis (LSH) [31]. According to the LCH, prior to hydrate formation, water molecules form labile, relatively long-lived, unstable clusters resembling the solid hydrate cavities. The forming structure is an agglomerate of cavities of various hydrate types [34,37]. Then, under favorable conditions, this agglomerate recombined into a solid clathrate hydrate phase [30], whose cage type ratio is closer to that of a hydrate crystal. This recombination, i.e., the formation of stable crystalline sI hydrate, passes through the polycrystal structure that consists of sI and sII nanocrystals connected by 5^12^6^3^ [34]. Conversely, LSH suggests that guest molecules are arranged in a configuration similar to that of the crystal due to concentration fluctuations, and then the water molecules rearrange around the guest molecules to form the hydrate structure proposed by Radhakrishnan and Trout [31]. The concept of LSH, in which the hydrate can appear first with the ordering of guest molecules followed by water molecules, differs from LCH. Moreover, the authors of LSH argue against LCH by showing that it is thermodynamically favorable for the labile clusters to disintegrate rather than agglomerate to form a larger cluster.

In the hydrate phase, the concentration of hydrophobic gas molecules such as methane is more than two orders of magnitude higher than the solubility of methane in liquid water [36]. In order to increase methane solubility, it is possible to lower the temperature and/or increase the pressure of the mixture. In one study [38], the authors examined metastable methane solubility using MD simulations. It was shown that to increase the methane concentration in the water phase by two orders of magnitude at a fixed temperature, it is also necessary to increase the pressure by more than two orders of magnitude. It follows from the results of another study [39], at *T* = 258.5 K and *P* = 1.2 bar, the equilibrium concentration of methane in water is *x_M_* = 0.0005 molar fraction, and at *P* = 945 bar, *x_M_* = 0.05. In that study, the critical solubility, above which metastable solutions spontaneously form a hydrate phase, was found to be about 0.05 molar fraction, comparable to the concentration of methane in the hydrate (*x_M_* = 0.07) with half-filling of all the hydrate cavities [39].

The very first stage of clathrate hydrate formation is characterized by gas consumption into the water phase with the formation of metastable supersaturated water. This stage is often called the induction time. Increasing gas concentration is observed until the critical nuclei of a new hydrate phase are formed. A critical nucleus can be defined as the minimum amount of a new phase that is capable of existing independently [39]. In water + gas systems, relaxation processes involving the structural redistribution of dissolved guest molecules occur during the induction time, leading to the initial stages of growth phase formation. The metastable supersaturation phase remains a key step toward a complete understanding of hydrate nucleation [40].

The direct production of a supersaturated aqueous hydrate gas solution is still a challenge. However, indirect methods of obtaining supersaturated solutions do exist. Thus, for example, solid amorphous solutions of N_2_, O_2_, CO, and Ar have been obtained [41,42], from which gas hydrates have been produced subsequently at unexpectedly low external pressures.

A study of the initial steps of hydrate formation [43] showed that a dodecahedral water cage (DWC) immersed in the bulk water phase can efficiently adsorb dissolved methane molecules, which causes the DWC lifetime to increase exponentially with the number of methane molecules adsorbed. At the same time, adsorbed methane molecules reduce the number of hydrogen bonds that tend to break the DWC and change the hydrogen-bond network surrounding the DWC from a liquid water structure toward a cage-like hydrate structure. Such behavior argues in favor of considering hydrate nucleation as a cooperative process, where the guest molecules can stabilize the hydrate cages as well as the hydrate structure itself [15,43]. Another approach to describe the initial evolution of water + gas systems toward hydrate formation was reported by Guo et al. [44], who proposed the cage adsorption hypothesis (CAH). This theory predicts a critical concentration of ~0.04 molar fraction for methane aqueous solution, above which methane hydrate begins to nucleate [39].

Several MD techniques and methods have been developed to obtain deeper insight into the mechanisms behind the initial stages of the metastable supersaturation phase and the growth of the hydrate phase under a high driving force (high excess pressure and low temperature supercooling). The key parameters found are order parameters [45,46]. Nucleation pathways associated with both host and guest ordering and the mutual influence of guest and host ordering are being explored. The resulting parameters are used to describe the dynamics and mechanisms of hydrate formation and other key properties.

A number of different simulation studies [47,48,49,50,51] have revealed that nano-bubbles in the gas phase trapped inside the water phase could act as centers of hydrate nucleation at the liquid-gas interface that function as heterogeneous nucleation centers. Additionally, studies from the Uchida group [52,53] have shown that the induction time of C_2_H_6_ and Xe hydrates is significantly lower if the water solution is made using a hydrate with nano-/micro- bubbles that act as nucleation centers.

To produce hydrate structure formation in the presence of a gas-water interface, MD simulation studies are carried out at high pressures (from 500 to 2000 bar) and/or low temperatures [14,15,16,17,18,19,23,32,33,34,38,39,54,55,56,57,58]. In the referenced works, the nucleation time varied from several hundreds of nanoseconds up to several microseconds depending on the exact thermodynamic (*P*-*T*) conditions, selection of guest type, and water molecule model.

A different approach was applied by Sarupria and Debeneditti [59] to study a supersaturated homogeneous aqueous methane solution. The authors simulated a methane solution with a 0.07 mol fraction of methane at *T* = 240 K and *P* = 200 bar and reported a several hundred nanosecond nucleation time for a 0.07 molar concentration of methane and found structures similar to those reported in the case of a gas-water interface at similar temperature [14,15,16,17,18,19,32,33,34,38,39]. The nucleation time is drastically reduced at higher methane concentrations (0.08) [60]. Moreover, increasing the degree of subcooling leads to the formation of more ordered hydrate structures. Thus, homogeneous supersaturated methane aqueous solutions make it possible to model hydrate growth under low overpressure or undercooling [21,60,61] in contrast to models with separated phases.

Later simulations showed four phases of nucleation [62]. There is an increase in solvated methane concentration in the aqueous domain via diffusion through the methane−water interface, the formation of unstructured clusters of methane molecules solvated in water, decreasing the water content of the nucleus to a value compatible with the type II methane clathrate hydrate composition, and the final reordering of solvated methane and water as in the ‘blob’ hypothesis [34].

The structures of methane, propane, and methane + propane hydrates formed from water mixtures with high structural order and low computational time have been obtained using MD simulations [60,63]. In these works, a moderate subcooling level has been applied, and the overall thermodynamic conditions considered are within the *P* and *T* range of practical interest. Other important steps to revealing the hydrate formation mechanism were made by other studies [35,55,56,59,64], where the initial stages of the metastable supersaturation phase and growth of the hydrate phase were studied at high excess pressure and low temperature supercooling with an aqueous phase prepared by melting the hydrate phase. The study of hydrate nucleation in a “methane + water + THF” system [65,66] showed the dependence of induction time, growth rate, and cage structure on the *P*-*T* conditions and THF concentration.

We later studied a dynamic hydrogen-bond network in the liquid water phase [5,67]. Based on this, we developed efficient methods to analyze hydrogen-bond networks and a method for determining the hydrate nucleus. In previous work [68], we showed the possibility of methane hydrate growth from a bulk water + methane solution or in the presence of sea salt and hydrate seed at excess pressure without notable induction time. However, the nucleation mechanism was not studied.

In the present work, we have focused on an examination of the initial stage of the methane hydrate phase growth mechanism from a supersaturated methane solution without excess pressure and supercooling applied using MD simulation methods. The initial solutions are based on a uniform distribution of gas molecules inside the bulk water phase, which results in an ideally stirred solution. The aim is to study the dynamics of hydrogen-bond network reconfiguration and the influence of fluctuating hydrate nuclei on the hydrate growth process in supersaturated methane solution, with an emphasis on studying the behavior of water molecules with a high degree of local order at the earliest steps.

## 2. Results and Discussion

### 2.1. Fluctuation Character of the H-Bond Network Structure at the Hydrate-like Structure Pre-Nucleation Interval

The significant effect of guest molecules on the properties of the H-bond network, leading to the formation of small 5^12^ fluctuation cavities at the initial stage of the hydrate-like structure (amorphous hydrate) nucleation process in a methane solution at low pressure, has been shown. 

The characteristic time dependence of the number of large (5^12^6^2^) and small (5^12^) sI hydrate cavities in solution with *C* = 0.6 and *P* = 1 bar is presented in Figure 1 for a single trajectory at which the formation and dissipation of sI cavities were observed over the time period 0–10 ns. The simulation could be divided into several intervals according to the features of small and large cavity formation. In the first interval (up to 1 ns) no cavities were formed. The second interval, 1–4.5 ns, is characterized by the formation and dissipation of single hydrate cavities (fluctuation cavities) in water solution (see Figure 1b), which have lifetimes of about one picosecond and could be the predecessors of hydrate nuclei. In the next interval (4.5–6 ns), at least one cavity exists in the volume, but because of fluctuations, it is possible to observe up to three different separate small cavities (see Figure 1c). Then after 6 ns, transformation of the hydrogen-bond network is observed, which leads to the formation of 2-3-4 small cavities as shown in Figure 1d, and only after a simulation time of 7 ns, does the fluctuation of large sI cavities become detectable and then persistent after 7.5 ns, although their number fluctuates (see Figure 1e). After a 10 ns simulation time, steady growth of the amorphous hydrate is observed. The number of cavities grows, and fluctuations do not lead to the disappearance of a significant fraction of the cavities. During the modeling of system transformations, there is no particular spatial ordering of the cavities, and the entire hydrogen-bond network of water molecules is involved. In this simulation, we focused only on the small (5^12^) and large (5^12^6^2^) cavities that form the sI hydrate, but as noted earlier, other cavity types are present in this system and show similar behavior.

Figure 2a,b shows the time dependence of the order parameters *F*_3_ and *F*_4_ in the first 10 ns of simulation and comparisons to the bulk hydrate, bulk water, and cavity-forming water molecules in solution. The difference between *F*_3_ and *F*_4_ for bulk water and *F*_3*_liquid*_ and *F*_4*_liquid*_ is due to the fact that the TIP4P model series does not have an H-O-H angle equal to the right tetrahedral angle of 109.47 degrees, which is used in some other potentials. The value of the order parameters for bulk water does not change over time because the water is in an equilibrium state. The sharp decrease in the *F*_3_ parameter (an increase in tetrahedrality compared to pure water) occurs during the first 1 ns, which shows steady H-bond network reorganization (the difference is ~0.01 and gradually increases). Qualitatively similar results are presented for the *F*_4_ local torsion angle order parameter: whole solution order is close to pure water; however, the order parameters of cavity forming molecules in solution for both *F*_3_ and *F*_4_ are very close to those of a crystal in the same temperature and pressure region. Thus, the scattered cavities formed in the H-bond network are topologically close to hydrates. In general, the presence of dissolved methane molecules has an ordering effect on the structure of water. This water structure ordering in the presence of high concentrations of guest molecules is similar to the behavior of water molecules described in [69], where clathrate-like structures were shown to form under ambient conditions in the presence of confinement and a hydrophobic environment. The authors noted that as hydrophobic compounds, many different molecules can be chosen, such as alkanes such as methane, ethane, propane, etc. In this case, one may see the earliest stages of hydrate formation [34,62], which can be caused by a high local concentration of methane and an ordered arrangement of gas molecules [70], which is initially ensured by the homogeneity of the solution. Unlike many other works, here the distributed volume cavity formation can be observed. The reason for this is connected with the collective rearrangement of solution water molecules bonded by the hydrogen-bond network, which is discussed next.

Figure 2c shows the time dependence of the portion of highly ordered water molecules for which the parameter *F*_3_ is lower than 0.025 (highly ordered water molecules, HOWMs). This value was chosen because the water molecules forming the cavities have an average value of *F*_3_ < 0.025. As expected, the order parameters of these molecules are much more “crystalline” than for other molecules. The increase in the number of HOWMs compared to bulk water confirms the ordering of the structure prior to the formation of regular hydrate cavities. The fluctuations in the number of HOWMs indicate the dynamic nature of the local environment of all molecules united by the H-bond network.

Figure 3 shows the visualization of HOWMs during the first nanoseconds. At *t* = 0 ns (Figure 3a), the number of HOWMs is insignificant; however, at *t* = 0.1 ns (Figure 3b), this number is about ~20–25% of all molecules. It is important to note that HOWMs are delocalized in this time period (Figure 3a–f) in the entire system volume and do not form a single local cluster. This allows the cavity to fluctuate and form with any group of H-bond network HOWMs.

### 2.2. Radial Distribution Function of HOWMs

The radial distribution functions (RDF) were calculated to characterize the reorganization of the H-bond network structure. Figure 4a–d shows the oxygen–oxygen RDF for HOWMs in methane solution, bulk water, and bulk hydrate systems. The solution model peaks transform from liquid type with three peaks to hydrate type with four peaks over the time period. The position of the third peak shifts closer to the hydrate type. The high value of the first peak for the solution could indicate that HOWMs form groups that are uniformly distributed and form hydrate-like structures/fragments. At a larger time scale, further convergence of the RDF lines for the HOWMs and the general structure of water can be expected. The gas–oxygen RDF is discussed in the Appendix A and is presented in Appendix A.

### 2.3. Growth Outside the Stability Region

Using the MD method, we examined the evolution of two sets of independently generated water + methane systems with different methane concentrations in the system at *P* = 1 bar and *T* = 270 K. The total number of water molecules was 10,001 and 1200 (for *C* = 0.6-small). The chosen concentrations are much higher than the equilibrium methane concentration in water, and thus the supersaturated homogeneous solutions obtained here are metastable. The pressure and temperature fluctuations are presented in Appendix A. The simulation for *C* = 0.2 over 100 ns (Figure 5a–d) shows neither hydrate-like structure formation nor phase separation but a metastable supersaturated solution with homogeneously distributed gas molecules. At methane concentrations *C* = 0.4 (Figure 5e–h) and *C* = 0.6 (Figure 5i–l), the simulation results in the appearance of a volume region with a different methane concentration. The region with higher methane concentrations corresponds to a hydrate-like structure phase. The other region is a water solution with a greatly decreased methane concentration in comparison with the initial supersaturated phase. Increasing concentration (*C* = 0.8, Figure 5m–p) leads to the appearance of the third region of high methane concentration—gas nanobubbles. The formation of a gas phase at *C* = 0.8 is a competing process with the amorphous hydrate formation process: unlike the other systems where all runs provide similar results, for *C* = 0.8 in the first run, behavior similar to *C* = 1.0 is observed, whereas in the second run we observed a hydrate-like structure growth process but with a nano-bubble formation. At *C* = 1.0 (Figure 5q–t), no hydrate-like structure formation was observed.

In the simulations of the small system for 700 ns, a similar mechanism of hydrate formation was observed. Longer simulation times showed a significant decrease in the gas concentration in solution and the transition of the gas to the amorphous hydrate phase (Figure 5u–x). In this case, a fluctuation rearrangement of the amorphous hydrate structure is observed, which, however, qualitatively does not undergo significant changes with further modeling. Figure shows the separated amorphous hydrate layer and liquid solution layer. Thus, the same system evolution path at different sizes can be seen and will be discussed later.

Complete 100 ns evolutions of metastable supersaturated solutions with *C* = 0.2, 0.4, 0.6, 0.8, and 1.0 are presented in Appendix A, respectively. The structures bonded by stable hydrogen bonds and incorporated into these structures can be seen at early stages that indicates the presence of “blobs” that is one step in the way towards hydrate formation [34]. According to a previous study [38], nanobubbles formed in the water phase at *C* = 0.8 and *C* = 1.0 could be considered as hydrate nucleation centers at the microsecond timescale [71].

Experimentally obtaining supersaturated liquid solutions at atmospheric pressure is a task that does not have a direct solution, but the formation of a solid solution seems to be possible through the deposition of water vapor and methane on a cooled substrate [41,42].

The chemical potential of water molecules in the hydrate (or hydrate-like structure) phase depends not only on the interaction between water molecules in the hydrate lattice but also on the number of guest molecules in the hydrate cavities and exists as an entropy contribution, as shown in our previous work [72]. The value of this contribution at a certain guest molecule concentration exceeds the difference between the chemical potential of water molecules in the liquid phase and the empty hydrate lattice framework. In this case, due to a sufficiently large number of dissolved methane molecules, the filling of the hydrate-like structures is large enough to make it energetically preferable with respect to the liquid phase even at atmospheric pressure. We observed this phenomenon at concentrations of 0.4 (6.5 mol%) and 0.6 (9.45 mol%) during MD simulations. The formation of cavities may be preceded by the distribution and concentration of methane molecules that provoke the growth of hydrate, as stated in [31,70]. In this case, the formation of an amorphous hydrate is observed, similar to the works [34,37], but with a difference, which lies in the delocalization and the formation of cavities in the entire system volume. In the case presented here, the systems go through the stages of ‘blob’ and amorphous hydrate formation, which could be followed by the formation of crystalline hydrate.

The molecular dynamics method is not designed to study thermodynamic stability; the focus is on studying the dynamics of system change [73], such as viscosity, diffusion, and other transport properties. Therefore, the study of thermodynamic stability requires the use of other methods or techniques, especially in the region close to phase equilibrium, where the influence of fluctuations cannot be neglected.

In molecular dynamics simulations, the characteristic cell size rarely exceeds 100 Å, that is quite sufficient to describe the phenomena of interest. Nevertheless, the formation of hydrate nanocrystals with a characteristic size of ~200 Å, which can be in an aqueous gas solution without causing crystallization of the solution, has been experimentally shown [74]. This once again confirms the unpredictability of the behavior of small systems, which is somewhat similar to manifestations of the confinement effect. The authors of [75] showed that nanoporous media make it possible to increase gas solubility/uptake under milder conditions. So, it was shown that the confinement effect makes it possible to obtain methane hydrate at a lower pressure [76] and even at 1 bar [77].

When the concentration increases to more than 0.8 (12.2 mol%), phase separation occurs, which is in agreement with theoretical work [60]. This leads to a significant decrease in methane concentration in the water phase and disruption of this solid structure formation mechanism. With a decrease to *C* = 0.2 (3.36 mol%) or lower methane concentration, the amount of dissolved methane molecules in the water phase is insufficient for an adequately large entropy contribution to the water chemical potential, which was confirmed by the presence of a minimum methane concentration in the solution to initiate any solid structure formation [39].

Molecular dynamics simulations using the TIP4P/Ice water model [55] showed the growth of methane hydrate at *T* = 245 K and *P* = 0.2 bar, which is outside the methane hydrate stability area [78]. The authors hypothesized that unexpected hydrate growth is due to the spherical shape of the initially separated gas phase, which can create additional pressure and cause stable hydrate growth [55].

Previous results [21,60,61] assume that using a supersaturated methane solution as the initial structure allows one to observe hydrate formation at moderate conditions such as lower pressure and/or lower sub-cooling. Our results show that the effect of supersaturation of an aqueous solution of methane has greater importance than was previously suggested and allows hydrate-like structure formation at 1 bar.

Figure 6a shows the smoothed dependence of the potential energy of systems with different contents of guest molecules depending on the simulation time. It can be seen that for a system with a concentration *C* = 0.2, the energy value does not change with time, which indicates the absence of any evolution of the system. However, at higher concentrations (*C* = 0.4, 0.6), a monotonic decrease in potential energy is observed, which is accompanied by a rearrangement of the hydrogen-bond network of the system into an amorphous hydrate. At a concentration of *C* = 0.8, the behavior of the potential energy indicates two parallel processes: the process of rearranging the network of hydrogen bonds into a hydrate-like structure at the initial stage of simulation (up to 20 ns) and the process of isolating guest molecules into a separate phase, which remains the only one after 20 ns of simulation. A further increase in the concentration of guest molecules (*C* = 1.0) leads to the immediate release of the gas phase of guest molecules, which is expressed in an increase in potential energy. These graphs show the advantage of the formation of hydrate-like structures at certain concentrations of methane from the point of view of potential energy.

The construction of a solution model with a concentration of guest molecules significantly exceeding the thermodynamically stable one creates the conditions for the formation of amorphous hydrate. It can be assumed that while the concentration of guest molecules in the liquid phase of water is high, the process of hydrate formation will continue. However, as the guest molecules in the liquid phase are exhausted, the driving force of hydrate formation caused by their presence will gradually decrease, which will subsequently lead, first, to a slowdown in growth and then to the beginning of its dissociation. In our modeling, we observe only the initial stage of this process, where only amorphous hydrate growth is observed. With continued modeling, one may expect “amorphous hydrate—methane solution” phase separation (as it was shown in Figure 5u–x) and the formation of hydrate, whose structure will be closer to crystalline, taking into account amorphous cavities on its surface. Although the decomposition of hydrate-like structures is eventually expected due to thermodynamic conditions, it can be seen that the resulting structures are quite viable, as shown by the modeling of a small system (Figure 5u–x). The stability of the detected phases and their evolution with time are not fully studied here and can be the subject of a separate study.

Figure 6b shows the dependencies of the total number of 5^12^, 5^12^6^2^, 5^12^6^3^, and 5^12^6^4^ hydrate, and 4^1^5^10^6^2^, 4^1^5^10^6^3^, and 4^1^5^10^6^4^ hydrate-like cavities, forming an amorphous hydrate agglomeration, in the systems. A continuing increase in the number of cavities is seen for systems *C* = 0.4 and 0.6. For the *C* = 0.4 system, about ~13% of methane molecules are captured by cavities of various types during the formation of amorphous hydrate, and further growth can be expected. In the case of the system *C* = 0.6, by the end of the first 100 ns of simulation, more than 25% of methane molecules are captured. More molecules could be captured by unfinished cavities (i.e., cavity caps). Further, the rate of cavity formation slows down significantly, which is consistent with the fact that the formation of cavities at *C* = 0.2 is not observed in the considered time interval, and the composition of the liquid phase of the solution is still far from equilibrium. Due to phase separation, no growth is observed in the system *C* = 0.8 after 50 ns of simulation; however, the decay (dissociation) of the formed structure is also not observed. The increase in potential energy shown in Figure 6a for this system is due to the continued release of methane from solution.

The volume distribution of hydrate cavities for the cavity forming systems *C* = 0.4, 0.6, 0.8 is presented at Figure 7. Initial steps of delocalized hydrate formation in the entire system volume are clearly observed. Fluctuation formations and the decay of cavities were observed throughout the simulation time. Due to the relatively low methane concentration at *C* = 0.4, some hydrate nuclei are dissipating or merging into the larger one. Higher methane concentrations (*C* = 0.6 and 0.8) lead to more sustainable growth in the number of nucleation centers. As a result, the formation of an amorphous hydrate can be seen. The decrease in cavity number for *C* = 0.8 that may look like localization is associated with the release of gas from the solution shown at Figure 5m–p. The figures clearly show that the cavities in the resulting structures are connected to each other to form an amorphous structure. So, it is not possible to distinguish a clear crystal structure of sI, sII, or other types on the given time scale. For the sake of clarity, the 4^1^5^10^6^2^, 4^1^5^10^6^3^, and 4^1^5^10^6^4^ cavities were not shown; however, these cavities are directly connected with the presented cavities, and due to their topology, they can easily transform to the hydrate type.

The dependence of the number of cavities on time for systems with different methane concentrations is presented in Figure 8. Methane forms an sI hydrate that consists only of 5^12^ (Figure 8a) and 5^12^6^2^ (Figure 8b) cavities. Nevertheless, during the structure formation process, other cavity types could be formed because of fluctuations, surface effects, and other effects. These cavity types were also drawn in the plots (Figure 8c–g). For systems with concentrations of *C* = 0.4, 0.6, and 0.8, the number of observed cavities increases with time. It should be noted that the formation of large (5^12^6^2^) cavities occurs after the onset of the formation of small (5^12^) ones. An increase in the number of different types of cavities is strongly correlated with methane concentrations in the water phase. The system with *C* = 0.6 shows the fastest speed, and, therefore, this concentration is the most favorable for hydrate-like structure formation under these conditions. We believe that at different pressure/temperature conditions, the most favorable methane concentration should be nearly the same. In other words, amorphous hydrate formation is faster if the gas/water molar relation is within a certain range. Similar dependencies for series A and several smaller models are given in the Appendix A.

The sI hydrate unit cell consists of six large and two small cavities, but in our calculations, this relation is not met due to a large number of other cavity types (5^12^6^3^, 5^12^6^4^, 4^1^5^10^6^2^, 4^1^5^10^6^3^, and 4^1^5^10^6^4^) that reconfigure in later formation phases. Thus, the 4^1^5^10^6^2^ cavity is topologically very close to the large 5^12^6^2^ cavity present in the sI hydrate and can easily reconfigure into a proper large 5^12^6^2^ cavity. Using such an assumption, the resulting relationship is not far from the target 3/1 value. Disintegration (change of structure/size) of the cavities is observed only as fluctuations in time. In the time interval of 0–100 ns, growth of the amorphous hydrate (*C* = 0.4 and 0.6) was observed. The case of *C* = 0.8 is exceptional, where slow amorphous hydrate decomposition is caused by gas phase separation. However, the resulting phases (0.4 and 0.6) are not thermodynamically stable, but under the given *p-T* conditions, they are metastable. Additional driving forces (such as supercooling, high pressure, promoters, or a helper gas) could be used to stabilize the structure. 

The dependence of the normalized H-bond number on time and concentration is shown in Figure 9a and confirms previous results. The growth of *N_H-bond_/N_mol_* indicates an increasing number of molecules involved in the formation of four H-bonds that are connected with crystal-like ordering, where each molecule forms exactly two donor and two acceptor H-bonds (*N_H-bond_*/*N_mol_* = 2). At *C* = 0.2, the local structure of the solution does not change in terms of mean values. The decrease in *N_H-bond_*/*N_mol_* at *C* = 1.0 clearly indicates that a high proportion of molecules form the phase boundary. The *N_H-bond_*/*N_mol_* steadily increases for solutions with initial *C* = 0.4 and 0.6, and in the case of *C* = 0.8, during the first 25 ns (amorphous hydrate growth). In this case, the growth rate of the hydrate-like structure from solution with *C* = 0.8 is higher than at 0.6 and much higher than at *C* = 0.4.

The distribution of average *N_H-bond_/N_mo_*_l_ in dependence on methane concentration makes (Figure 9b) it possible to estimate the methane concentration boundaries for successful hydrate-like structure formation. At *C* > 0.2 and *C* < 0.8 a stable increase in the *N_H-bond_/N_mol_* number is observed. Thus, this range could be used for hydrate-like structure formation at low pressure. 

The *F*_3_ and *F*_4_ order parameters were calculated for a 0–100 ns time interval (Figure 9c,d, respectively) for the considered solutions. Solutions with *C* = 0.4 and 0.6 show an increase in the gradual ordering of the H-bond network during the entire simulation time. The difference from the ideal value is due to the fact that a significant fraction of water remained in the liquid phase. The solution with *C* = 0.8 revealed ordering until *t* = 23 ns; however, the number of hydrate cavities increased. The discrepancy between these results is explained by an increase in methane nano-bubbles and the influence of the bubble surface. The disordering of an H-bond network solution with *C* = 1.0 caused by methane bubble formation is expressed as an increase in the *F*_3_ parameter and a decrease in the *F*_4_ parameter. The solution with *C* = 0.2 shows no significant change over time. The value *F*_4_ ≈ –0.03 for *C* = 0.2 and 1.0 corresponds to a liquid (bulk water), and the small difference from *F*_4*_liquid*_ is not significant and can be caused by the features of the geometry of the chosen water model.

The obtained results indicate the cooperative nature of the hydrate-like structure nucleation process, which is determined by the rearrangement of the hydrogen-bond network, which is associated with an individual change in the tetrahedral order parameter of each water molecule in solution.

A change in the distribution of gas molecules also indicates a partial transition of the solution to the amorphous hydrate. Due to the initial uniform mixing of gas molecules, the peaks in the regions of 3–5 Å (first peak) and 6–8 Å (second peak) are present (Figure 10a–e) at *t* = 1 ns. The comparison with bulk hydrate and the increase in the first and second peaks of *g(r)* with time, as well as the appearance of the third peak, clearly confirm amorphous hydrate growth in solutions with *C* = 0.4 (Figure 10b), 0.6 (Figure 10c), and 0.8 (Figure 10d). The value of *g(r)* for the second peak exceeds the height of the first peak (0.4 and 0.6) or reaches a close value (0.8); however, in the case of 0.4 and 0.6, the distribution becomes constant with distance, indicating the presence of a significant amount of methane mixed with water even after *t* = 100 ns. However, in a system with a methane concentration of 0.8, due to the formation of a bubble (phase separation), the distribution value does not reach a constant value with increasing gas-gas distance. Moreover, at *t* = 20 ns, the growth of the second peak is insignificant. This is in agreement with theoretical works on carbon dioxide [79] and methane [62] hydrate formation.

The second peak of *g(r)* for the *C* = 1.0 solution (Figure 10e) smoothens over 20 ns, which is faster than the disordering of order parameters taking place during 30–40 ns and indicates the separation of phases is nearly complete, but residual methane segregates slowly. A decrease in *g(r)* with r clearly confirms the phase separation. 

For the solution with *C* = 0.2 (Figure 10a), where no hydrate growth or phase separation is observed over 100 ns, no qualitative differences appear over time. The constant value of *g(r)* at high *r* values indicates that the methane concentration is not high enough to break the H-bond network.

## 3. Calculation Methods

Five homogeneous methane + water mixture models with different compositions (see Table 1) were constructed. The selected methane concentrations (*C*) correspond to 20%, 40%, 60%, 80%, and 100% rates of single filling of small (5^12^) and large (5^12^6^2^) cavities of sI hydrate composed of 10,001 and 1200 (*C* = 0.6-small) water molecules. In contrast with previous works [35,55,56,59,64], where the aqueous phase was (homogeneously) supersaturated with gas and prepared by melting the hydrate phase, in the current study the initial supersaturated solution was similar to a different study [60]. The water molecules were placed at the nodes of a virtual cubic lattice with a step of 3 Å and a random orientation in space. The number of lattice sites slightly exceeded the number of water molecules, so some sites remained empty, but the model remained cubic. The gas molecules were also randomly arranged at the nodes of another independent virtual lattice with the same step size. For verification purposes, two series of simulations (A and B) were carried out with different distributions of water and gas molecules. By default, we present the results corresponding to series B, and the qualitative differences between series A and B are described. Before initiating the simulation, molecular non-overlap was confirmed.

All simulations were conducted at *T* = 270 K and *P* = 1 bar in the *NPT*-ensemble with a 1 fs timestep using LAMMPS software [80]. The time constants of the Nose–Hoover thermostat and barostat [81,82] implemented in LAMMPS were 300 fs and 3000 fs, respectively. Details of selected parameters are presented in the Appendix A. Long-range interaction was calculated using the PPPM method [83]. The cut-off distance of the Coulomb and van der Waals forces was set to 13 Å. The SHAKE algorithm [84] was used in order to maintain the geometry of the water molecules. Periodic boundary conditions were applied for every model.

Snapshots for analysis purposes were written every 100 ps for all long-term simulations (more than 10 ns). In order to study the pre-nucleation processes and early steps of hydrate-like structure nucleation itself for the large 0.6 model (10,001 H_2_O + 1043 CH_4_) during the first 10 ns, snapshots were taken every 100 fs. In order to reveal structure changes due to nucleation, the pure water model containing 10,001 H_2_O molecules and the bulk methane hydrate model containing 6 × 6 × 6 sI unit cells (9936 H_2_O + 1728 CH_4_) were simulated at the same conditions over 10 ns and 1 ns, respectively.

The TIP4P/Ice potential [85] (*σ* = 3.1668 Å, *ε* = 0.21095 kcal/mol, *q_O_* = −1.1794|*e*|, *q_H_* = +0.5897|*e*|, O-M distance = 0.1577 Å, O-H distance = 0.9572 Å, and H-O-H angle = 104.52°) was used to describe the water molecules due to its high melting point accuracy in comparison to other simple 3-point and 4-point potentials. Water molecules were considered rigid. Methane molecules were considered single-point particles, with potential parameters described elsewhere [86].

The structures of the homogeneous mixture (series A and B, series of smaller systems) and the simulation run parameters with force-field data are presented in Appendix A.

The analysis of the hydrogen-bond (H-bond) network was carried out as described in our previous works [5,67]. The geometric criterion for H-bonding was chosen as follows: *R_OO_* ≤ 3.2 Å and *∠HOH* ≤ 30°. The local (short-range order) tetrahedrality order parameter *F*_3_ [45] and torsion angle order parameter *F*_4_ [46] were used in this study. Reference points of the *F*_3_ parameter for the ideal water crystal and bulk liquid are *F_3_crystal_* = 0 and *F*_3*_liquid*_ ≈ 0.1. Reference points of the F_4_ parameter for the hydrate, bulk liquid, and ice are *F*_4*_hydrate*_ ≈ 0.7, *F*_4*_liquid*_ ≈ 0, and *F*_4*_ice*_ ≈ −0.3.

To find the cavities in a water solution, a graph of the connectivity of water molecules by H-bonds was created based on the geometric criterion of H-bonding. The closed chains of 4, 5, and 6 H_2_O molecules were found using this graph. These chains were considered the faces of a polygon. Subsequently, a connection graph of the faces was created through their common (adjacent) sides. Using this graph, the closed polyhedral structures were searched, among which sI hydrate cavities (5^12^ and 5^12^6^2^) and topologically close hydrate ones 4^1^5^10^6^2^, 4^1^5^10^6^3^, and 4^1^5^10^6^4^ cavities were selected and studied. The obtained sets of coordinates of water molecules were visualized.

## 4. Conclusions

The collective nature of the methane hydrate nucleation process in the dynamic H-bond network of TIP4P/Ice water outside the region of thermodynamic stability was considered. At *T* = 270 K and *P* = 1 bar, the process of amorphous hydrate formation depended on the methane concentration in water was observed. At methane concentrations of 0.4 (6.5 mol%) and 0.6 (9.45 mol%), growth of the amorphous hydrate phase was observed, which showed the intriguing behavior of methane-containing hydrate-like structure formation outside the region of thermodynamic stability of methane hydrate. At a concentration of 0.8 (12.2 mol%), the competing process of the formation of methane bubbles in water was found. Early stages of hydrate formation are observed: ordering of the hydrogen-bond network, formation of ‘blobs’ throughout the entire solution volume, and amorphous hydrate. It can be assumed that the nature of this process is connected with the sufficiently high entropy contribution of the guest molecules to the chemical potential of the water molecules in the hydrate phase. This contribution allows the transformation of the H-bond network of a metastable liquid solution into the H-bond network of a more energetically preferable amorphous hydrate structure. The further behavior of obtained hydrate-like structures needs additional research at different thermodynamic conditions, however, the small system simulation showed the dynamic amorphous hydrate with a certain number of hydrate cavities exists at the microsecond time scale.

The collective character of H-bond network reorganization in methane solution was revealed as an increase in water short-range order tetrahedrality at the initial stage of simulation (0–1 ns) before the appearance of the first fluctuation of sI hydrate cavities. Molecules with a high degree of tetrahedrality (low *F*_3_ order parameter values) are distributed uniformly. A significant change in *F*_3_ preceded the formation of cavities, and thus the entire network of hydrogen bonds was involved in the amorphous hydrate formation process. Fluctuating cavities appeared in the interval 1.5–6 ns. Later, the fluctuating and stable cavities were constantly present in the system, and an increase in their number indicated the initial stage of the formation of nanohydrate phases.

The formation of hydrate cavities and hydrate nanostructures under the described conditions is possible using real-time Raman spectrometry as in Ref. [87].

## Figures and Tables

**Figure 1 molecules-28-02960-f001:**
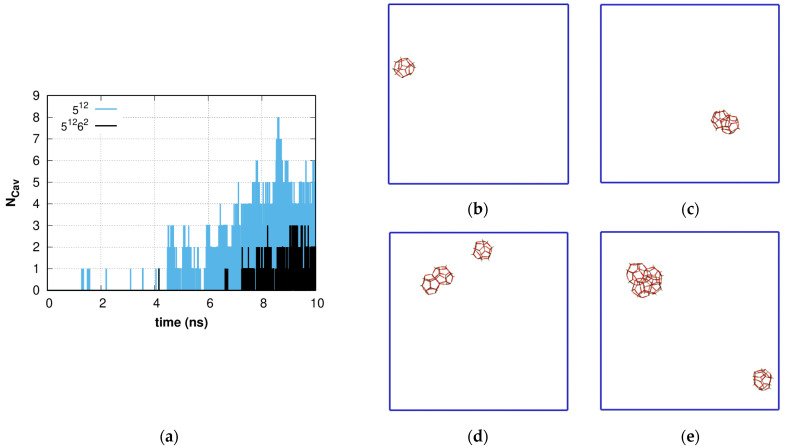
(**a**) Time dependence of proper large and small sI cavity number. Spatial distribution of small and large cavities at different simulation time points: (**b**) *t* = 1.500 ns; (**c**) *t* = 4.651 ns; (**d**) *t* = 8.365 ns; (**e**) *t* = 10.0 ns. Intermediate time steps are presented in Appendix A.

**Figure 2 molecules-28-02960-f002:**
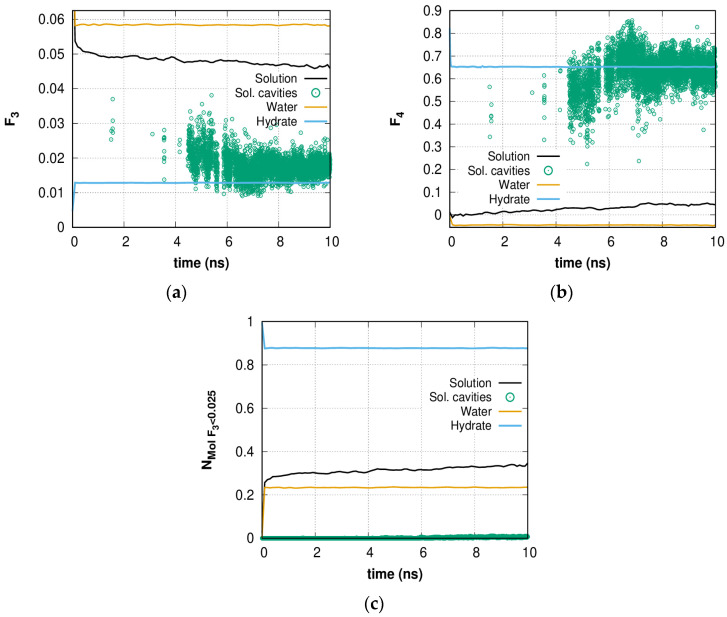
Time dependence of (**a**) *F*_3_ and (**b**) *F*_4_ order parameters of 0.6 H_2_O + CH_4_ solution (*black line*), pure water (*orange line*), and bulk hydrate (*blue line*) phases (time period corresponding to 0–1 ns and stretched to 0–10 ns for comparison). The green open circles represent the corresponding data for cavity-forming molecules. The data are smoothed; (**c**) Time dependence of the portion of all molecules having molecular parameter *F*_3_ < 0.025 in 0.6 methane solution, bulk hydrate, and pure water. The data are smoothed.

**Figure 3 molecules-28-02960-f003:**
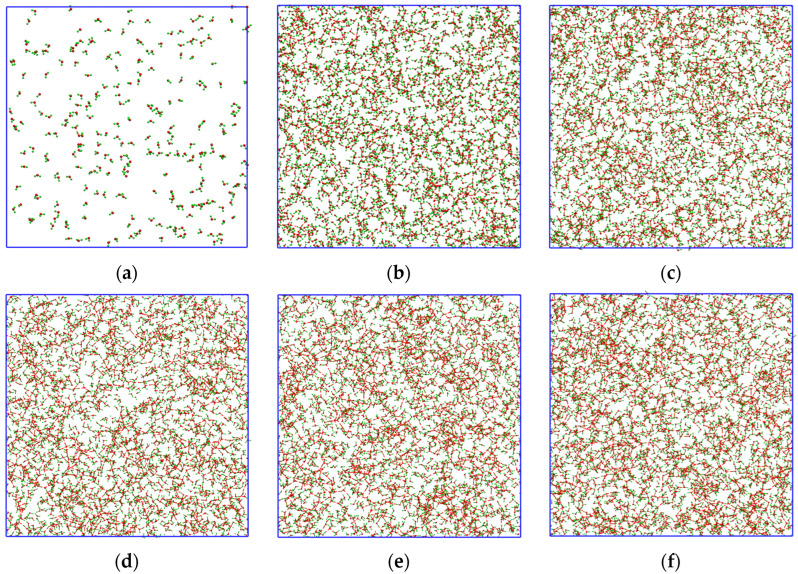
Spatial distribution of highly ordered water molecules (HOWMs) in 0.6 solution at different simulation time points: (**a**) *t* = 0.0 ns; (**b**) *t* = 0.1 ns; (**c**) *t* = 1.0 ns; (**d**) *t* = 3.0 ns; (**e**) *t* = 6.0 ns; (**f**) *t* = 10.0 ns.

**Figure 4 molecules-28-02960-f004:**
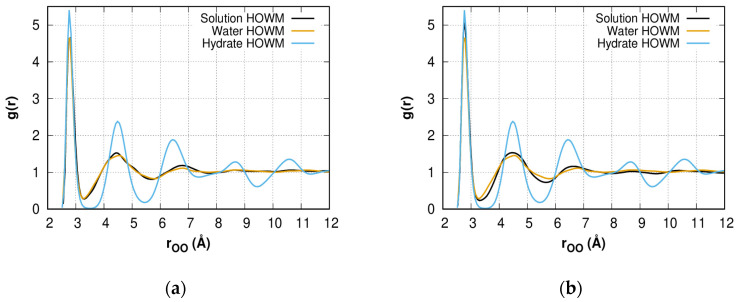
The highly ordered water molecule (HOWM) oxygen–oxygen (eigen *F*_3_ < 0.025) RDF *g(r)* for the 0.6 methane solution, pure water, and bulk phase methane hydrate at different simulation time points: (**a**) *t* = 0.1 ns; (**b**) *t* = 4.0 ns; (**c**) *t* = 6.0 ns; (**d**) *t* = 10.0 ns. Water and hydrate data are averaged over the simulation time. The *r_OO_* distance distribution step is 0.01 Å.

**Figure 5 molecules-28-02960-f005:**
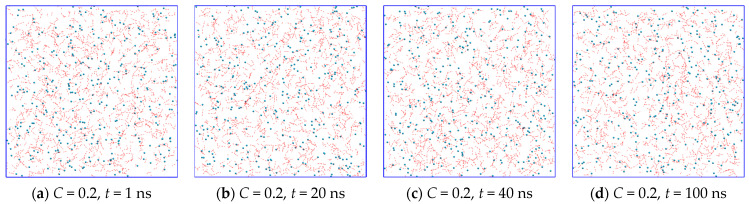
Spatial distribution of H-bonds and gas molecules in the methane + water solution at concentrations of (**a**–**d**) *C* = 0.2, (**e**–**h**) *C* = 0.4, (**i**–**l**) *C* = 0.6, (**m**–**p**) *C* = 0.8 and (**q**–**t**) *C* = 1.0 at simulation time points: 1 ns, 20 ns, 40 ns, and 100 ns and (**u**–**x**) for replicated 3 × 1 × 1 times *C* = 0.6-small at time points 1, 50, 100, and 700 ns. Coordinates of water molecules are averaged over 1 ns in order to show the stable hydrogen bonds (red dashed lines). Hydrate-like structure growth at *C* = 0.8 is not characteristic.

**Figure 6 molecules-28-02960-f006:**
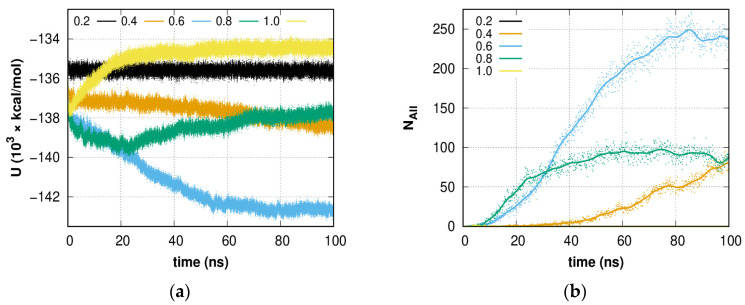
Time dependence of (**a**) system potential energy and (**b**) overall cavity number. *Solid lines* represent the smoothed data of the cavity number.

**Figure 7 molecules-28-02960-f007:**
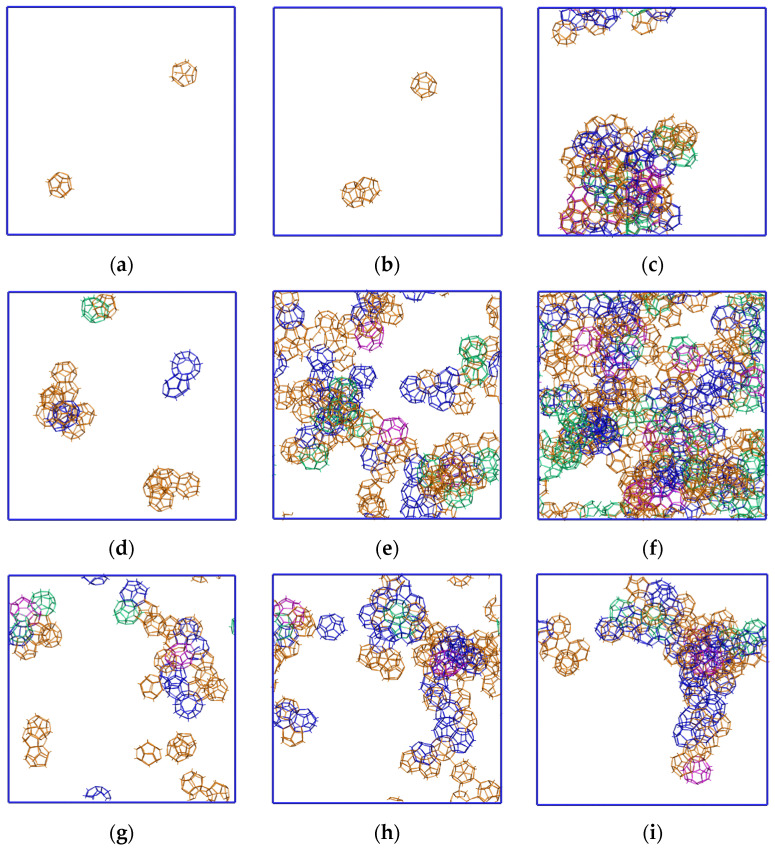
Spatial distribution of 5^12^ (*orange*), 5^12^6^2^ (*blue*), 5^12^6^3^ (*green*), and 5^12^6^4^ (*magenta*) cavities for (**a**–**c**) *C* = 0.4, (**d**–**f**) *C* = 0.6, and (**g**–**i**) *C* = 0.8 systems at simulation time points: 20 ns, 40 ns, and 100 ns. Hydrate-like structure growth at a concentration of 0.8 is not characteristic; the drawing angle is the same as in Figure 5.

**Figure 8 molecules-28-02960-f008:**
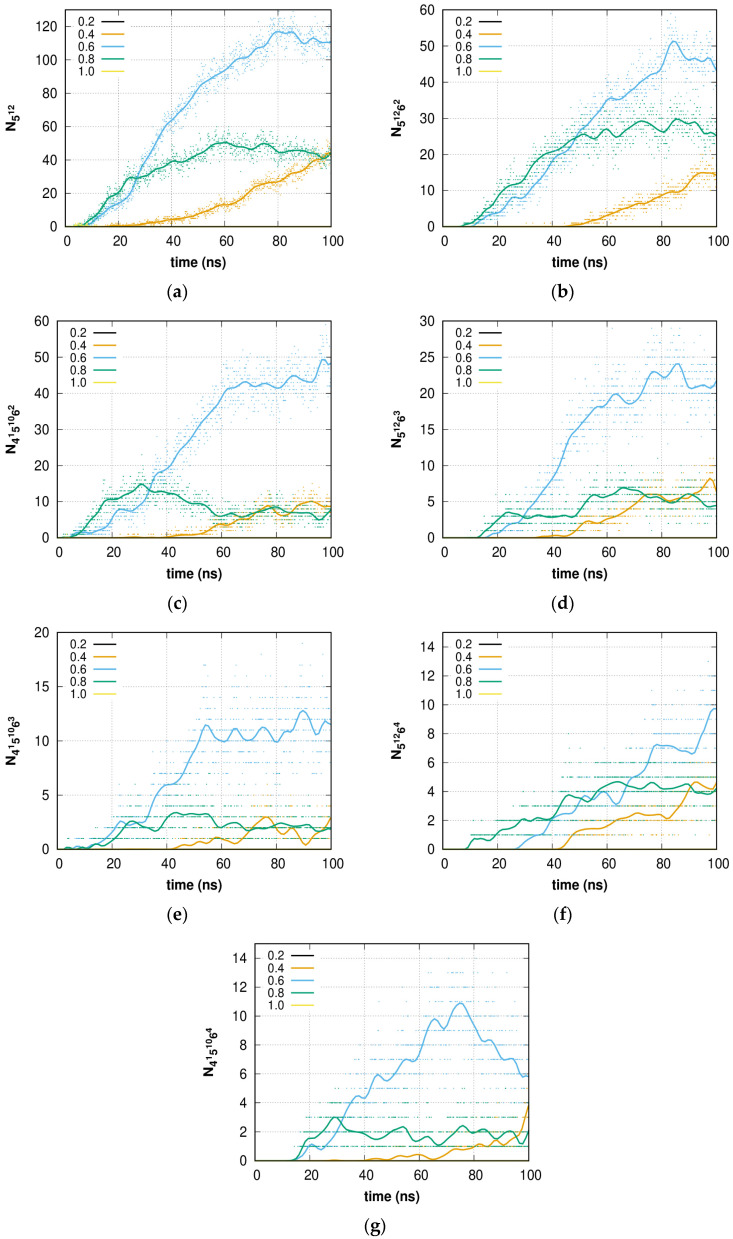
Time dependence of the number of (**a**) 5^12^; (**b**) 5^12^6^2^; (**c**) 4^1^5^10^6^2^; (**d**) 5^12^6^3^; (**e**) 4^1^5^10^6^3^; (**f**) 5^12^6^4^; (**g**) 4^1^5^10^6^4^ type cavities of amorphous methane hydrate agglomeration. *Solid lines* represent the smoothed data.

**Figure 9 molecules-28-02960-f009:**
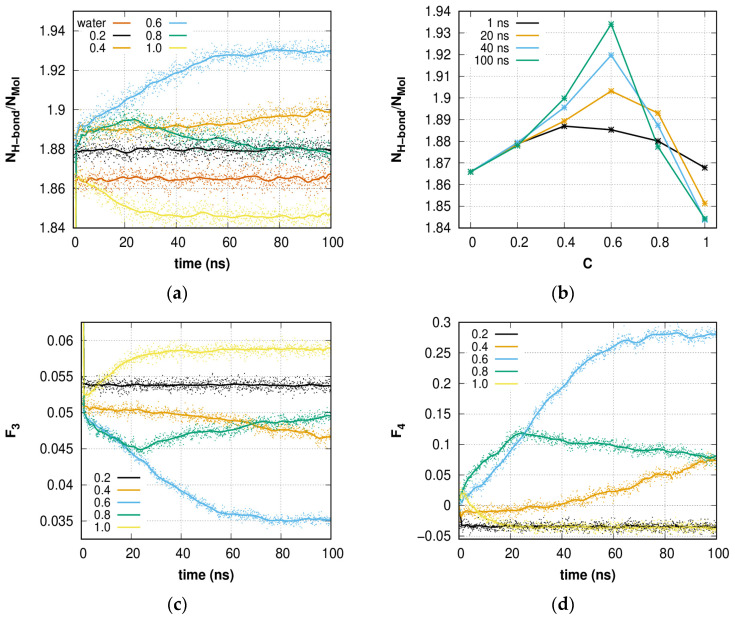
(**a**) Time dependence of H-bond number per water molecule and (**b**) normalized H-bond number as a function of gas concentration at different time points. Time dependence of (**c**) *F*_3_ and (**d**) *F*_4_ order parameters. When *t* = 0 ns, *F*_3_ = 0.1 for all methane concentrations. *Solid lines* at (**a**,**c**,**d**) represent the smoothed data.

**Figure 10 molecules-28-02960-f010:**
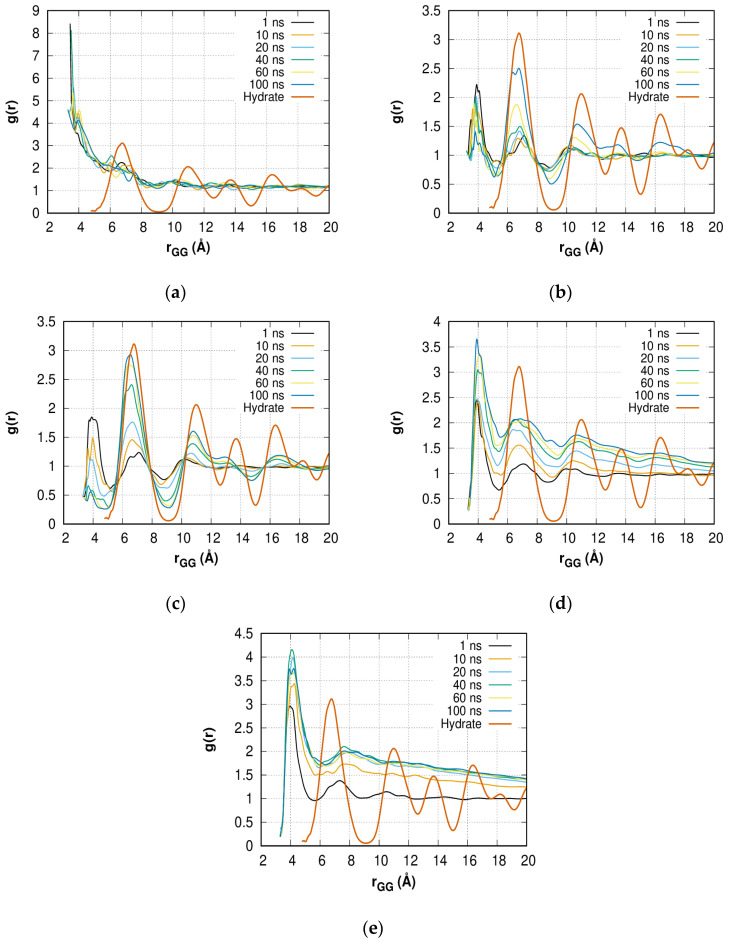
Gas–gas RDFs *g(r)* in (**a**) *C* = 0.2; (**b**) *C* = 0.4; (**c**) *C* = 0.6; (**d**) *C* = 0.8; (**e**) *C* = 1.0 solutions at different time points in comparison with the bulk hydrate phase. The *r_GG_* distance distribution step is 0.01 Å.

**Table 1 molecules-28-02960-t001:** Designation of concentration, *C*, for methane (*N_M_*) + water (*N_W_*) solution compositions and correspondence to molar concentration (*x_M_*).

*C*	*N_M_*	*N_W_*	*x_M_*
0.2	348	10,001	3.36 mol%
0.4	696	10,001	6.5 mol%
0.6	1043	10,001	9.45 mol%
0.8	1391	10,001	12.2 mol%
1.0	1739	10,001	14.8 mol%
0.6-small	125	1200	9.43 mol%

## Data Availability

The datasets generated and analyzed during the current study are available from the corresponding author by request.

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
