# Peer review of "Molecular Dynamics Study of Clathrate-like Ordering of Water in Supersaturated Methane Solution at Low Pressure"

_molecules, 2023, doi:10.3390/molecules28072960_

Round 1
Reviewer 1 Report
The manuscript is well-organized and can be directly accepted for publication.
1) Figure 1 needs improvement.
2) In Figure 2 and 8, the font can be larger.
Author Response
Reviewer #1
The manuscript is well-organized and can be directly accepted for publication.
Q1) Figure 1 needs improvement.
Our Reply: The legend of Fig. 1a was modified by increasing space between lines. The contrast of cavities was improved in Figs. 1b-e.
Q2) In Figure 2 and 8, the font can be larger.
Our Reply: It was corrected. In Figures, the font was increased in the revised manuscript.
Reviewer 2 Report
This seems to me an excellent work that provides valuable information for a better understanding of the nucleation and propagation of gas hydrates.
In the typescript with title Molecular dynamics study of clathrate-like ordering of water in supersaturated methane solution at low pressure, the authors use classical molecular dynamics to discern the mechanism for the growth of methane hydrates from a supersaturated aqueous solution, a topic of high interest in various fields, mainly in the extraction and exploitation of gas from marine subsoil. That, apart from the inherent scientific merit of reliably describing nucleation phenomena. They made a thorough revision of the state-of-the-art literature and convincingly guide the reader to fathom the relevance of their work. The methods are well described; the force-fields used for the intermolecular interactions, TIP4P/Ice for water, and a single site for methane, are among the best in reproducing the ices and gas-hydrates. The analyses made of the raw data are sound and support the conclusions, especially the proposal that hydrate formation is connected with a sufficiently high entropy contribution of the guest molecules to the chemical potential of the water molecules in the hydrate phase.
Besides recommending the publication of the article in its present form, I suggest increasing the sizes of the texts and numbers in Figs. 2, 8 and 10.
Author Response
Reviewer #2
This seems to me an excellent work that provides valuable information for a better understanding of the nucleation and propagation of gas hydrates.
In the typescript with title Molecular dynamics study of clathrate-like ordering of water in supersaturated methane solution at low pressure, the authors use classical molecular dynamics to discern the mechanism for the growth of methane hydrates from a supersaturated aqueous solution, a topic of high interest in various fields, mainly in the extraction and exploitation of gas from marine subsoil. That, apart from the inherent scientific merit of reliably describing nucleation phenomena. They made a thorough revision of the state-of-the-art literature and convincingly guide the reader to fathom the relevance of their work. The methods are well described; the force-fields used for the intermolecular interactions, TIP4P/Ice for water, and a single site for methane, are among the best in reproducing the ices and gas-hydrates. The analyses made of the raw data are sound and support the conclusions, especially the proposal that hydrate formation is connected with a sufficiently high entropy contribution of the guest molecules to the chemical potential of the water molecules in the hydrate phase.
Q1) Besides recommending the publication of the article in its present form, I suggest increasing the sizes of the texts and numbers in Figs. 2, 8 and 10.
Our Reply: It was corrected. The texts and numbers were increased in Figures of the revised manuscript.
Reviewer 3 Report
The presentation is detailed and very well written. The methods are appropriate, rigorously applied and discussed in full.
My only request is that the authors re-work some of the figures to make more readable when reduced. For example, the text/legend on Figs 2, 8 and 10 are too small. Figure 1(a) has overlapping text.
Otherwise, nice work.
Author Response
The presentation is detailed and very well written. The methods are appropriate, rigorously applied and discussed in full.
Q1) My only request is that the authors re-work some of the figures to make more readable when reduced. For example, the text/legend on Figs 2, 8 and 10 are too small.
Our Reply: It was corrected. The texts and numbers were increased in Figures of the revised manuscript.
Q2) Figure 1(a) has overlapping text.
Our Reply: It was corrected. The legend of Fig. 1a was modified by increasing space between lines.
Reviewer 4 Report
The manuscript of Belosludov et al reported the results of MD simulations of clathrate-like ordering of water molecules in supersaturated methane solutions at different CH4 concentrations and conducted at low pressure. The authors have use a common simulation software and suitable potentials for CH4 and water, including ice structure-suitable one. The properties that have been evaluated in this research are the cavity number, energy vs. time, various order parameters vs. time, the radial distribution function for the characterization of the water ordering in phases used as descriptor of clathrate formation, visual inspection of the structuring, hydrogen bonding in the system, which are again the standard for the evaluated systems.
In general, this manuscript fits the scope of the journal Molecules MDPI, reports interesting and new results to the problem of gas solubility in water with following water structures study at lower temperatures and normal pressure.
However, prior the final decision is made concerning the manuscript, its content should be adjusted and improved. In the following, the authors will see the list of questions and suggestions, which to my opinion will be helpful for the revision of the text.
1. The introduction is lengthy and unstructured. I would suggest the authors reconsidering the part describing the results of somebody's simulations. I do not see the structure of this description - for the moment it looks like description of the separated facts/findings without a purpose. E.g. why the information about thermostat is important here (lines 154-158)? Thermostats in MD dissipate the excess of energies of the processes happening in the simulation cells, during hydration, or structuring, or isomerisation. Why the authors think, this information is important here to mention? Another point is the problem of hydrophobic hydration, not named like this, but in fact meant in lines 68-82. Hydrophovic hydration is important, but why it is mentioned in the paper about clathrates? This type of hydration can occur without damaging water HB network, especially for small solutes (smaller than 1 nm in diameter), which contradicts to the text on lines 68-70. Please adjust the introduction accordingly and answer the questions.
2. lines 104-106 are unclear, please correct the text.
3. Why particle-particle particle-mesh Ewald is important for the simulation of the systems without ions?
4. How the analysis and visualization of cavities, polygons have been performed?
5. only static descriptors are considered by the authors (unless they are calculated versus time). Why do not include the dynamic properties of water during clathrate formation, and near a hydrophobic methane? For example, water retardation, velocity auto-correlation functions, Coefficients of translational diffusion?
6. Do the authors see the second hydration shell around methane molecules? how water is structured there?
7. Some plots should be substantially enlarged - Figures 2, 8, 10 - this is a requirement for the revised version.
8. In conclusions you should shortly explain the thesis you mentioned in the abstract - about "intriguing behavior"... of solutions with 6.5 and 9.45 mol% methane concentration.
9. Please illustrate the "blob"; if it is seen in your snapshots, please make a comment there. also, unfortunate to use this term since it can be mixed with analogous one used for years in polymer physics (see early papers by A.R. Khokhlov and A.Yu. Grosberg).
Author Response
The manuscript of Belosludov et al reported the results of MD simulations of clathrate-like ordering of water molecules in supersaturated methane solutions at different CH4 concentrations and conducted at low pressure. The authors have use a common simulation software and suitable potentials for CH4 and water, including ice structure-suitable one. The properties that have been evaluated in this research are the cavity number, energy vs. time, various order parameters vs. time, the radial distribution function for the characterization of the water ordering in phases used as descriptor of clathrate formation, visual inspection of the structuring, hydrogen bonding in the system, which are again the standard for the evaluated systems.
In general, this manuscript fits the scope of the journal Molecules MDPI, reports interesting and new results to the problem of gas solubility in water with following water structures study at lower temperatures and normal pressure.
However, prior the final decision is made concerning the manuscript, its content should be adjusted and improved. In the following, the authors will see the list of questions and suggestions, which to my opinion will be helpful for the revision of the text.
Q1) The introduction is lengthy and unstructured. I would suggest the authors reconsidering the part describing the results of somebody's simulations. I do not see the structure of this description - for the moment it looks like description of the separated facts/findings without a purpose.
Q1a)E.g. why the information about thermostat is important here (lines 154-158)? Thermostats in MD dissipate the excess of energies of the processes happening in the simulation cells, during hydration, or structuring, or isomerisation. Why the authors think, this information is important here to mention?
Our Reply: In the initial version of the article, the mention of a thermostat was added due to the fact that we considered it important to point out the possibility of using a thermostat in the study of hydrate formation, since in a fairly large amount of work an ensemble is used that does not use a thermostat, for example, NVE. We agree with the reviewer that this is not an important detail to include in the introduction, so it has been removed.
Q1b)Another point is the problem of hydrophobic hydration, not named like this, but in fact meant in lines 68-82. Hydrophovic hydration is important, but why it is mentioned in the paper about clathrates? This type of hydration can occur without damaging water HB network, especially for small solutes (smaller than 1 nm in diameter), which contradicts to the text on lines 68-70.
Our Reply: Hydrophobic hydration is important for understanding the dissolution of hydrophobic molecules and the formation of clathrate hydrate. The solubility of hydrophobic molecules in water under normal conditions is very low and does not affect the structure of the solution as a whole, but changes it locally. For the formation of a hydrate, it is required to dissolve several orders of magnitude more molecules. In this case it is not a local effect: as a result of hydrophobic hydration, the entire structure of the solution is transformed. In the cited articles cited, the changes of the aqueous environment by non-polar molecules were considered. However, the network of hydrogen bonds was not directly studied, therefore, the mention of the network of hydrogen bonds was removed from this paragraph. It can be assume that the network may change, but not destroy or collapse. Our study shows that the formation of cavities within the network of hydrogen bonds is the result of the collective nature of this network. The relevant paragraph has been corrected (see Correction below).
Correction: We added in the Introduction (the paragraph 3 of the revised manuscript)
Hydrophobic hydration is important for understanding the dissolution of hydrophobic molecules and the formation of clathrate hydrate. The solubility of hydrophobic molecules in water under normal conditions is very low and does not affect the structure of the solution as a whole, but changes it locally. For the formation of a hydrate, it is required to dissolve several orders of magnitude more molecules. In this case it is not a local effect: as a result of hydrophobic hydration, the entire structure of the solution is transformed.
Q1c) Please adjust the introduction accordingly and answer the questions.
Our reply: “Introduction” section was modified accordingly.
Q2) lines 104-106 are unclear, please correct the text.
Our Reply: Two sentences (“It was shown that to increase methane concentration in the water phase by two orders in magnitude at fixed temperature.” and “It is also necessary to increase the pressure by more than two orders.”) is one sentence that was mistakenly divided into two ones.
Correction: It was shown that to increase methane concentration in the water phase by two orders in magnitude at fixed temperature it is also necessary to increase the pressure by more than two orders.
Q3) Why particle-particle particle-mesh Ewald is important for the simulation of the systems without ions?
Our Reply: Despite the fact that there are no ions in the studied systems (we have no considered seawater), the TIP4P/Ice model for water molecule has charges that are given in the fourth paragraph of the Calculation Methods section. This requires taking into account the Coulomb interaction. The PPPM method is the most optimal because it allows to simulate the large system with efficient speed of calculation.
Q4) How the analysis and visualization of cavities, polygons have been performed?
Our Reply: To find the cavities in water solution a graph of the connectivity of water molecules by H-bonds was created based on the geometric criterion of H-bonding. The closed chains of 4, 5, 6 H2O molecules were found using this graph. These chains were considered as faces of a polygons. Subsequently, a connection graph of the faces was created through common (adjacent) sides. Using this graph, the closed polyhedral structures were searched, among which sI hydrate cavities (512 and 51262) and topologically close to hydrate ones 4151062, 4151063, 4151064 cavities were selected and studied. The obtained sets of coordinates of water molecules were visualized.
This part was added at the end of the “Calculation Method” section.
Q5) Only static descriptors are considered by the authors (unless they are calculated versus time). Why do not include the dynamic properties of water during clathrate formation, and near a hydrophobic methane? For example, water retardation, velocity auto-correlation functions, Coefficients of translational diffusion?
Our Reply: For studying the dynamics of cavity formation, which is the main purpose of this work, the numbers of hydrate and hydrate-like cavities were calculated. In order to study the collective behavior of the hydrogen bond network, the order parameters (F3 and F4), as well as other characteristics were calculated. For these purpose, the water retardation, velocity auto-correlation functions, coefficients of translational diffusion is not necessary. However, we agree that these parameters by the reviewer are important for studying the formation of hydrates from separated phases.
Q6) Do the authors see the second hydration shell around methane molecules? how water is structured there?
Our Reply: We have not considered water molecules that form the second hydration shell. It requires the introduction of an additional criterion for the existence of such shell. We have considered only water molecules that form cavities (Fig. 1) and F3 and F4 order parameters (Fig. 2). This leads to the formation of clathrate-like structure (Fig. 7) due to grow of new cavities.
Q7) Some plots should be substantially enlarged - Figures 2, 8, 10 - this is a requirement for the revised version.
Our Reply: It was corrected.
Q8) In conclusions you should shortly explain the thesis you mentioned in the abstract - about "intriguing behavior"... of solutions with 6.5 and 9.45 mol% methane concentration.
Our Reply: The discovered "intriguing behavior" is that under certain conditions it is possible to observe the growth of methane-containing hydrate-like structures outside the region of thermodynamic stability of methane hydrate.
The corrected part is “At methane concentrations of 0.4 (6.5 mol%) and 0.6 (9.45 mol%), growth of the amorphous hydrate phase was observed that showed the intriguing behavior of methane-containing hydrate-like structures formation outside the region of thermodynamic stability of methane hydrate. At a concentration of 0.8 (12.2 mol%) the competing process of the formation of methane bubble in water was found.”
Q9) Please illustrate the “blob”; if it is seen in your snapshots, please make a comment there. also, unfortunate to use this term since it can be mixed with analogous one used for years in polymer physics (see early papers by A.R. Khokhlov and A.Yu. Grosberg).
Our Reply: The term “blobs” in our manuscript is a set of polyhedra formed by water molecules and enclosed gas molecules. The structure of such polyhedra may be far from the regular crystalline form of hydrate cavities. The term “blobs” was taken JACS, 2010, 132(33), 11806-11811 (ref. 34), whose theory of hydrate formation is often cited and has served as the basis for many papers. In view of the fact that the concept of “blobs” is widely used in theoretical works on hydrate formation, we think it is not necessary to provide additional clarifications in the context of studying the processes of clathrate hydrate nucleation. “Blobs” are generated in our systems. Some of them can be seen in Fig. 5 and on Supplementary Video. They can be characterized by increasing the number of hydrogen bonds and ordering of solutions prior to the active formation of polyhedral cages (Figs. 9a, c, d). However, the main interest for our work is the formation of regular hydrate cavities. Therefore, the separate illustration of is not required. The text has been corrected in term of using this term.
Correction:
- Inserted mention of “blobs” in the discussion of Fig.5, third paragraph of section “3.3.Growth outside the stability region”
The structures bonded by stable hydrogen bonds and incorporated into these structures can be seen at early stages that indicates the presence of “blobs” that is one step in the way towards hydrate formation [34].
- Remove the “blobs” in the discussion of Fig. 7 removed, because only crystalline cavities are shown in the figures and discussed in the text.
Round 2
Reviewer 4 Report
none